# 3D Cell Spheroids as a Tool for Evaluating the Effectiveness of Carbon Nanotubes as a Drug Delivery and Photothermal Therapy Agents

**Roman A. Anisimov [1], Dmitry A. Gorin [2] and Anatolii A. Abalymov [1,2,*]**

[1] Science Medical Center, Saratov State University, 410012 Saratov, Russia
[2] Center for Photonic Science and Engineering, Skolkovo Institute of Science and Technology, 121205 Moscow, Russia
[*] Correspondence: anatolii.abalymov@gmail.com

**Abstract:** Cell spheroids (CSs) are three-dimensional models in vitro that have a microenvironment similar to tissues. Such three-dimensional cellular structures are of great interest in the field of nano biomedical research, as they can simulate information about the characteristics of nanoparticles (NPs) by avoiding the use of laboratory animals. Due to the development of areas such as bioethics and tissue engineering, it is expected that the use of such 3D cell structures will become an even more valuable tool in the hands of researchers. We present an overview of carbon nanotubes (CNTs) research on CSs in order to determine the mechanism of their incorporation into CSs, drug delivery, and photothermal therapy. We will look at such areas as the application of CNTs for medical purposes, the advantages of spheroids over classical 2D cell culture, the ways in which CNTs pass into the intercellular space, and the ways in which they are absorbed by cells in a three-dimensional environment, the use of the spheroid model for such studies as drug delivery and photothermal therapy. Thus, CSs are suitable models for obtaining additional information on the required properties of CNTs in their application in nanobiomedicine.

**Keywords:** cell spheroids; carbon nanotubes; CNTs; nanomedicines; penetration

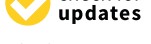



## 1. Introduction

The field of nanomedicine offers great opportunities for the development of new materials that can improve the therapy of various diseases [1,2]. Carbon nanotubes are fairly new nanomaterials that have unique properties and potential in various fields [3,4]. In particular, CNTs can broaden the horizon of biomedical research due to their important chemical, thermal, electrical, mechanical, and structural properties, which are currently of great interest. CNTs have a high modulus of elasticity and possess the properties of metallic, semiconducting, and superconducting materials [5]. Also, CNTs have a nanoarchitecture that allows both encapsulation of molecules inside and conjugation to the surface [6]. It has been shown that CNTs can be used in many applications, including biosensors [7,8], nanofluidic systems [9], biopharmaceutical applications [10], and diagnostic tools and devices in radiation oncology [11]. Unfortunately, CNTs still have no direct application in clinical settings due to the poor understanding of their biological properties and behavior in living objects [12]. In addition, in large-scale production, CNTs must also have well-characterized biological, environmental, and safety profiles. CNTs can vary significantly in size, morphology, structure, and purity depending on the method of preparation, purification, and functionalization used for their synthesis. Therefore, the interaction of CNTs with the biological environment is very complex and sometimes unpredictable, which requires an additional study on complex living systems [13].

For decades, monolayer cell cultures have been the primary model for the study of both molecules and NPs in the nanomedicine area [14–17]. This model has several ad-

vantages in the simplicity and accessibility of various analytical methods. However, 2D cell cultures do not reflect the full complexity of real tissues, although a huge number of drug-cell and nanoparticle-cell interactions have been obtained in 2D cultures [18]. One of the most important factors influencing the interaction between cells and NPs is cell–cell and cell–extracellular matrix (ECM) contacts in tissues, which supports homeostasis and specificity [19]. A three-dimensional cell model can mimic native tissue specificity better than cells cultured in monolayers by replicating such physiological interactions between cells and the extracellular matrix. Three-dimensional cell cultures are now used in a wide range of studies, including cell biology, tumor biology, epithelial morphogenesis, drug screening, and nanoparticle evaluation [20–24]. In addition, three-dimensional in vitro cellular systems have been used to reduce experimental uncertainties arising from various factors, such as pharmacokinetics and drug metabolism, in animal studies [25,26]. The complex three-dimensional network of the tumor microenvironment affects not only the penetration and distribution of therapeutic agents but also the function of many physiological factors [27]. Similarly, traditional animal tests often fail to predict the actual efficacy of a therapeutic agent in humans because animal cells, microenvironments, and physiology differ from human cells. This interspecies gap can be bridged by culturing human cells in 3D [28].

Here we discuss the current state of the art of 3D culture testing methods for recent developments in such areas of CNTs applications as drug delivery and photothermal therapy. On the one hand, an overview is given of the various modifications of CNTs that allow the particles to penetrate three-dimensional tissues, have good three-dimensional distribution, and act on them functionally. On the other hand, methods of investigating CNTs on three-dimensional CSs, the characteristics of the spheroids that affect the passage of particles into the intercellular space, and the absorption by cells, which directly affects the result of exposure, are discussed.

## 2. Properties, Modifications, and Application of CNTs

The nanoparticles made completely of carbon are known as carbon nanomaterials (CNMs). CNMs can be divided into 0D-CNMs (i.e., fullerenes, particulate diamonds, and carbon dots), 1D-CNMs (i.e., CNTs, carbon nanofibers (CNFs), and diamond nanorods), 2D-CNMs (i.e., graphene, graphite sheets, and diamond nanoplatelets), and 3D-CNMs. All decreased dimensionalities, including fullerenes, contain CNMs made completely of sp2-bonded graphitic carbon. All of the materials presented above can also be used for nano-biomedical applications, as evidenced by already existing scientific work [29–32].

Carbon nanotubes are an allotropic form of carbon. CNTs are well-ordered, high-aspect-ratio hollow graphite rods that were identified by Iijima in 1991 [33]. Since then, CNTs have been widely used in many areas, including electrode materials [34], nanoelectronics components [35], biosensors [36], strengthening of materials [37], and as components of biomaterials for drug delivery or other types of therapy [38–41]. The synthesis of CNTs is a broad topic and will not be described here in detail; however, it should be noted that the most used methods are electric-arc discharge [42], laser ablation [43], and the wide family of catalytic chemical vapor deposition (CCVD) methods [44]. A CNTs can be described as a rolled layer of graphene that can be opened and closed at the ends with fullerene caps [45].

One of the most important parameters of the CNTs is the number of concentric walls (Figure 1). The number of walls primarily determines the diameter of the CNTs. For example, single-walled carbon nanotubes (SWCNTs) have a small diameter (usually 1–2 nm), while multi-walled carbon nanotubes (MWCNTs) reach a diameter of up to 100 nm. Ref. [46] However, an increase in the number of walls also increases the number of defects, thus facilitating their modification and functionalization. Double-walled carbon nanotubes (DWCNTs) are located in the middle and are also quite promising since the diameter is still quite small, mechanical properties and electrical conductivity remain high due to the inner layer, but their surface modification is also possible due to the second wall [47].

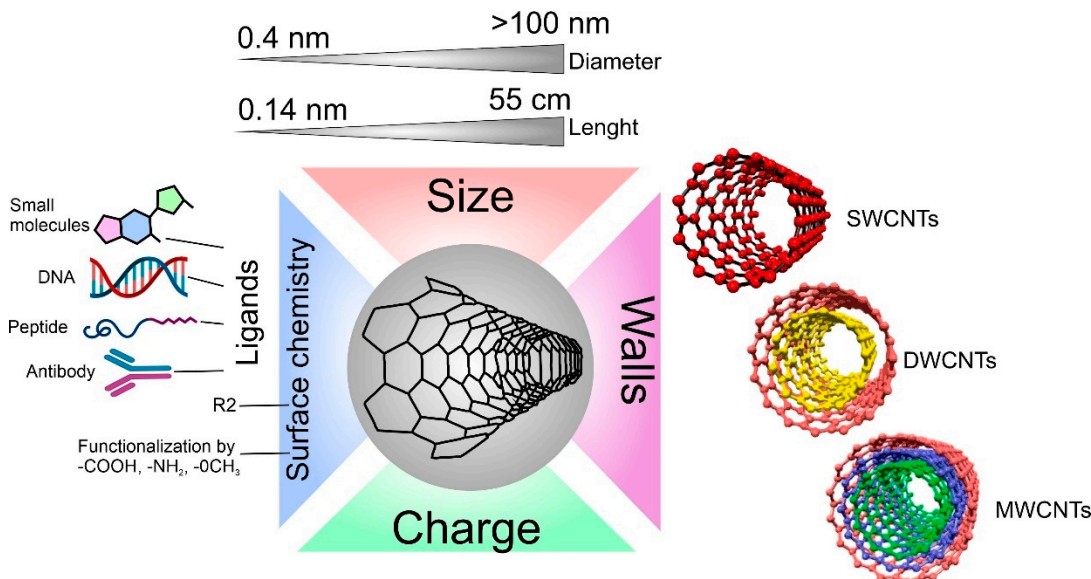

**Figure 1.** Carbon nanotube properties are important for interaction with cells.

The difference in the number of walls in CNTs can also affect cell viability in different ways [48]. For example, it has already been described that the difference in the cytotoxic effect on cells between single-walled and multi-walled CNTs is quite large [49]. It is hypothesized that MWCNTs lead to the production of reactive oxygen species (ROS), causing inflammation, while SWCNTs increase oxidative stress through damage to mitochondria [50]. Additionally, the difference in toxicity between multi-wall and single-wall CNTs is associated with their hydrophobic–hydrophobic interaction of MWCNTs with the cell membrane and following hole formation and loss of the plasma membrane integrity [51].

When synthesizing CNTs, parameters such as the diameter and length of the CNTs can be tuned; for example, it can be tuned by different flow rates and flow duration of the carbon precursor gas (C2H2) on the growth of CNTs by a thermal CVD method [52]. By changing the length of the carbon tubes, we change the specific area, which can be a very important parameter when using CNTs to load and deliver molecules [53]. However, the length and diameter of the CNTs also influence the degree of toxicity of the CNTs in vivo and in vitro [54]. It is proven that with an increase in the length of CNTs, the toxic effect also increases. This is because macrophages can more easily envelop CNTs with a shorter length [55,56].

Another key parameter, both for the physicochemical properties of CNTs and biocompatibility, is the surface chemistry of CNTs. Surface chemistry determines properties such as charge, hydrophobicity [57], photocatalytic activity [58], and the ability to bind to various biological molecules (one of the most important factors for the formation of a protein crown and connection with cells) [59]. Each of these factors can affect both in vitro co-localization and in vivo biodistribution [60,61]. One of the methods for functionalizing the surface of CNTs with groups is plasma treatment [62]. The advantage of plasma treatment is that it does not pollute the environment and provides a wide range of functional groups depending on the plasma parameters. Fine-tuning of the surface is achieved by changing the plasma processing parameters such as power, gases used, processing time, and gas pressure [63]. Surface functionalization of CNTs can provide good targeting to the desired cell type, such as surface functionalization with antibodies that selectively bind to the desired receptors (e.g., EGFR) on cancer cells. Such functionalization technologies are widely used in radioactivity and drug-delivery systems [64].

Depending on the properties of CNTs, they find various applications for biomedical purposes. Some of the most obvious applications of CNTs are molecule delivery [65], photothermal therapy [66], use as biosensors [67], and as a component for the synthesis

of hybrid materials for tissue engineering [68]. The choice of a molecule for delivery and its loading/conjugation primarily depends on the purpose of delivery. It can be peptides, nucleic acids, therapeutic molecules, etc.

Peptide delivery has already been demonstrated using the foot-and-mouth disease virus (FMDV) B-cell epitope, which was covalently bound to amino groups on the surface of CNTs. After conjugation, the peptides adopt a suitable secondary structure and can be recognized by specific monoclonal and polyclonal antibodies. Immunization of mice with FMDV peptide-nanotube conjugates induced a high humoral response compared to the free peptide. Similar results indicate the possibility of using carbon nanotubes as components for vaccines [69]. Delivery of nucleic acids using CNTs is also possible. This direction is extremely promising. For example, by functionalizing the surface with ammonium, nucleic acids bind to the surface of the CNT via electrostatic interaction [70]. The search for new and effective delivery systems for therapeutic agents also suggests the possibility of using CNTs as a carrier. Anti-cancer drugs such as doxorubicin (DOX) successfully bind to the surface of CNTs via π-π stacking, making the CNTs–DOX conjugation the basis of CNT-based drug delivery systems for the delivery of DOX to cancer cells [71].

CNTs can also be used for photothermal therapy, as they have excellent optical absorption in the visible and near-infrared sectors. When irradiated with near-infrared light, the local temperature of the tissues in which the CNTs are located rises to 40–45 °C and kills the cells that are within the heating radius [66]. Induction of high temperature for sufficient time causes physical damage such as protein denaturation and membrane lysis and can increase oxidative stress, eventually causing coagulative necrosis or apoptosis. The wide electromagnetic absorption spectrum of CNTs creates exceptional properties compared to other plasmon-heated nanomaterials (e.g., gold nanoshells and nanorods), which depend on the size and shape of CNTs [72]. Studies show that CNTs can achieve thermal destruction using tenfold-lower doses in solution and using threefold-lower laser power than that required for gold nanorods, and these also indicate that MWCNTs are more potent than bulk single-walled nanotubes in transferring the NIR light into heat.

Currently, the scientific community has identified three possible mechanisms of CNT cell toxicity. The first is based on irreparable mechanical damage to the membrane (cellular or nuclear) [73]. It is very likely that endocytosis, phagocytosis, or nanopermeasurement, which are the main ways in which the nanomaterial interacts with the lipid membrane, are strictly dependent on the geometry of the CNTs, especially their length [74]. The next putative mechanism of toxicity is oxidative stress, resulting from an increase in reactive oxygen species (ROS) and leading to numerous side effects in the cell, such as apoptosis, necrosis, cytochrome c release, oxidative DNA damage, reduced proliferation, inhibition of cell growth, etc. [73]. The last mechanism, the mechanism of genotoxicity, is in one way or another associated with DNA damage, characterized by a wide spectrum: CNTs interaction with proteins involved in chromosome aberration; CNTs effect on the mitotic spindle, micronuclei formation, indirect DNA oxidation, DNA breakage, etc. Although the toxic mechanisms of CNTs have been studied from several perspectives, there is still a strong correlation between triggered or inhibited molecular pathways and cell types [75]. Despite the described complexity of the processes occurring inside the cells targeted by CNTs, some scientific works suggest ways to overcome the toxic effects of CNTs by modifying the material surface with functionalizing groups, coating with metal oxides, or protein attachment. For example, coating with recombinant C1q, which is a protein that activates the classical pathway of the complement system involved in the innate immune system, is a promising approach to regulating inflammation. In addition, several theoretical studies on modeling a possible cellular response to CNTs demonstrate the mechanical interaction of nanotubes with the lipid layer or with proteins, suggesting a safer geometry of CNTs, which furthers the understanding of the action of CNTs on cells [76].

## 3. Properties, Fabrication, and Application of CSs

The use of cell cultures is the first step in biomaterial development, research, and clinical activities. There are a huge number of methods that are used to determine the cellular condition and behavior, for example, when exposed to cytostatics or on biomaterials surface [77]. However, in most scientific and research works for such tests, 2D cultures are used, i.e., cells located on the surface of the culture plastic and forming a monolayer [78]. However, we are in a three-dimensional world and consist of tissues, which in turn are also three-dimensional. Due to this three-dimensionality, tissues in the body have a large number of gradients, which can be mechanical [79], chemical [80], electrical [81], etc. [82]. Such gradients are practically impossible to obtain in 2D cultures, which makes them suitable but extremely distant from real biological objects. In this regard, the direction of studying 3D CSs has been actively developed in recent years [83]. Such 3D CSs can have properties much closer to real tissues and successfully fill the gap between cell culture and laboratory animals, which makes their use significant for biological research [84].

One important purpose of using an in vitro cell testing system is to replicate the cell microenvironment (Figure 2) [85]. For example, cells within a tissue are surrounded by neighboring cells and an extracellular matrix, which constantly provide the cells with biochemical and mechanical signals [86]. This 3D network of cell–cell and cell–ECM interactions maintains the specificity and homeostasis of a particular tissue [87]. As a result, testing and detection of interactions between NPs and 2D cultures cannot be called reliable since the tissue-specific properties characteristic of these cells in a 3D environment are lost. The complex 3D network of the tissue microenvironment influences not only the penetration and distribution of CNTs but also the function of many physiological factors [88]. In addition, conventional animal testing often fails to predict the actual efficacy of a therapeutic agent in humans because the cells, microenvironment, and physiology of animals differ from those of humans. This species gap can be bridged by culturing human cells in 3D. It is also necessary to understand that tissues have their specific properties, such as the size of the intercellular space [89], tissue stiffness [90], cell density [91], phagocytic function [92], and cell morphology [83]. As in solid tumors, cells in spheroids form layers; the outer layer consists of proliferating cells, followed by a layer of senescent cells. In the very center is the necrotic core. This gradient in cell survival and proliferation depends on the availability of nutrients and oxygen [93].

Intercellular contacts inside spheroids are much more complicated than in 2D cultures. Cells deposit ECM components such as collagen, laminin, fibronectin, proteoglycans, tenascin, etc. There are also a large number of intercellular compounds; for example, $\alpha5$- and $\beta1$-integrin, E-cadherins are a barrier to cytostatic molecules [94].

The spheroids can be formed from one or more cell types, such as breast cancer cells and fibroblasts, endothelial cells, and immune cells. In this way, cellular heterogeneity, which is present in normal and oncological tissues, can be achieved [95,96].

Optimizing spheroids for nanoparticle testing, in particular CNTs, is one of the important aspects of working with 3D cultures [97]. The cells used in the experiments should be in culture from 1 to 20 passages. The cells should be kept in a humidified incubator at 37 °C and 5% $CO_2$. Standard culture medium should be used for cultivation. Cells should have a 70–80% fill rate. Cultures should be transplanted with trypsin/EDTA solution (0.05% (wt/vol) trypsin and 0.02% (wt/vol) EDTA). There are several methods for creating cell spheroids, the hanging drop method [98], ultra non-adhesive well plates [98], magnetic nanoparticles [99], incubation in hydrogels [100], and the use of bioreactors. For CNTs testing, the first three methods are the most optimal since they are the easiest to use in all laboratories and have the smallest variation in the size of the spheroids. To create a spheroid 400 µm in diameter on the fourth day of formation, the desired concentration and cell proliferation rate must be determined. For this purpose, spheroids are formed from different numbers of cells (from 250 to 3000 cells/spheroid in the case of ultra non-adhesive plates, magnetic nanoparticles, and suspended droplet method). When creating spheroids, it is recommended to use a multichannel pipette, which will reduce the standard deviation

among the spheroid diameters to 5% in one plate and 10% in different experiments. Cells form a spheroid within 96 h in a $CO_2$ incubator at 37 °C. A phase-contrast microscope with 5x and 10x lenses is used to determine the size of the spheroid. The microscope is used to assess the integrity, diameter, volume and roundness of the spheroid. Once the optimal number of cells has been determined, it is possible to proceed with CNTs testing. This requires titration of CNTs and making 2x solutions of the substances tested. After that, 50 μL of the medium must be removed from the plate where the spheroids are formed, and 50 μL of the test solution must be added, thus making a concentration of CNTs of the desired concentration. The spheroids can then be incubated with the CNTs for the desired time.

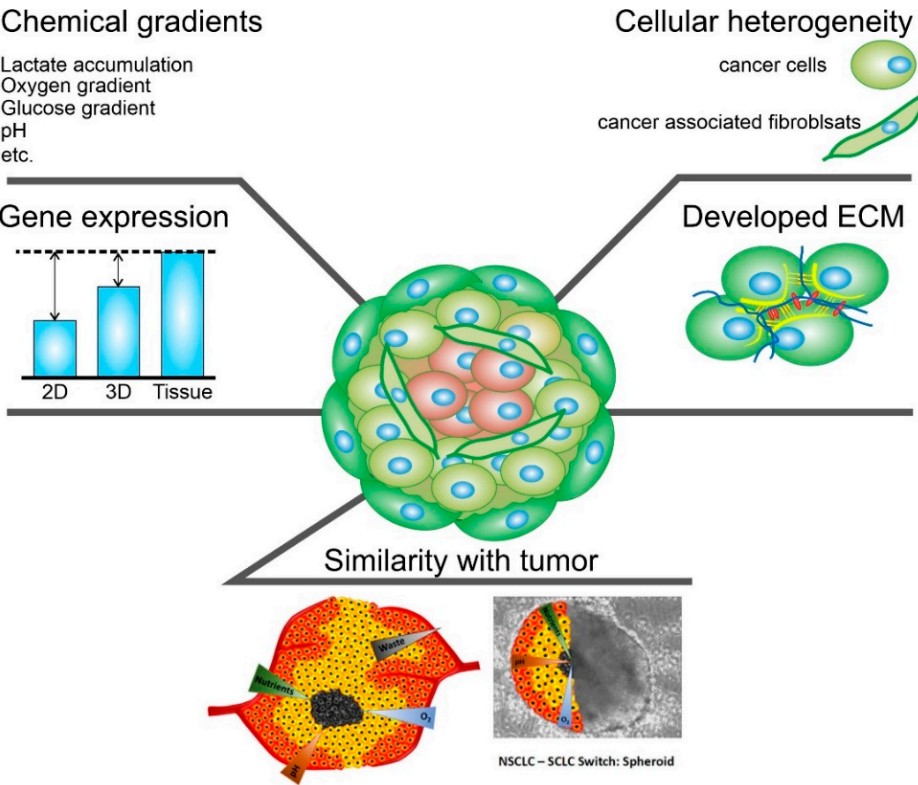

**Figure 2.** Schematic representation of the main characteristics of 3D spheroids that are crucial.

## 4. Mechanism of CNTs Uptake by Cells and Spheroids

The ECM that surrounds the cells serves as a good barrier to the penetration of therapeutic agents and NPs, including CNTs [24]. There are two types of transport of molecules and NPs into the spheroid: transcellular and diffusion through the extracellular matrix [101]. In the first case, the cells must absorb the carriers and pass them on to each other until the carrier reaches the cells of the necrotic nucleus. In the second case, CNTs must pass into the extracellular space, which usually has a size of 25–500 nm (Figure 3) [102]. In both cases, it depends on two parameters: the type of tissue and the properties of CNTs. It is worth paying attention not only to the properties of CNTs but also to the properties of other models NPs that have already been studied for penetration into tumor spheroids. The main properties of nano- and micro-sized objects that can be absorbed by cells are their size, shape, charge, surface chemistry, and rigidity [103].

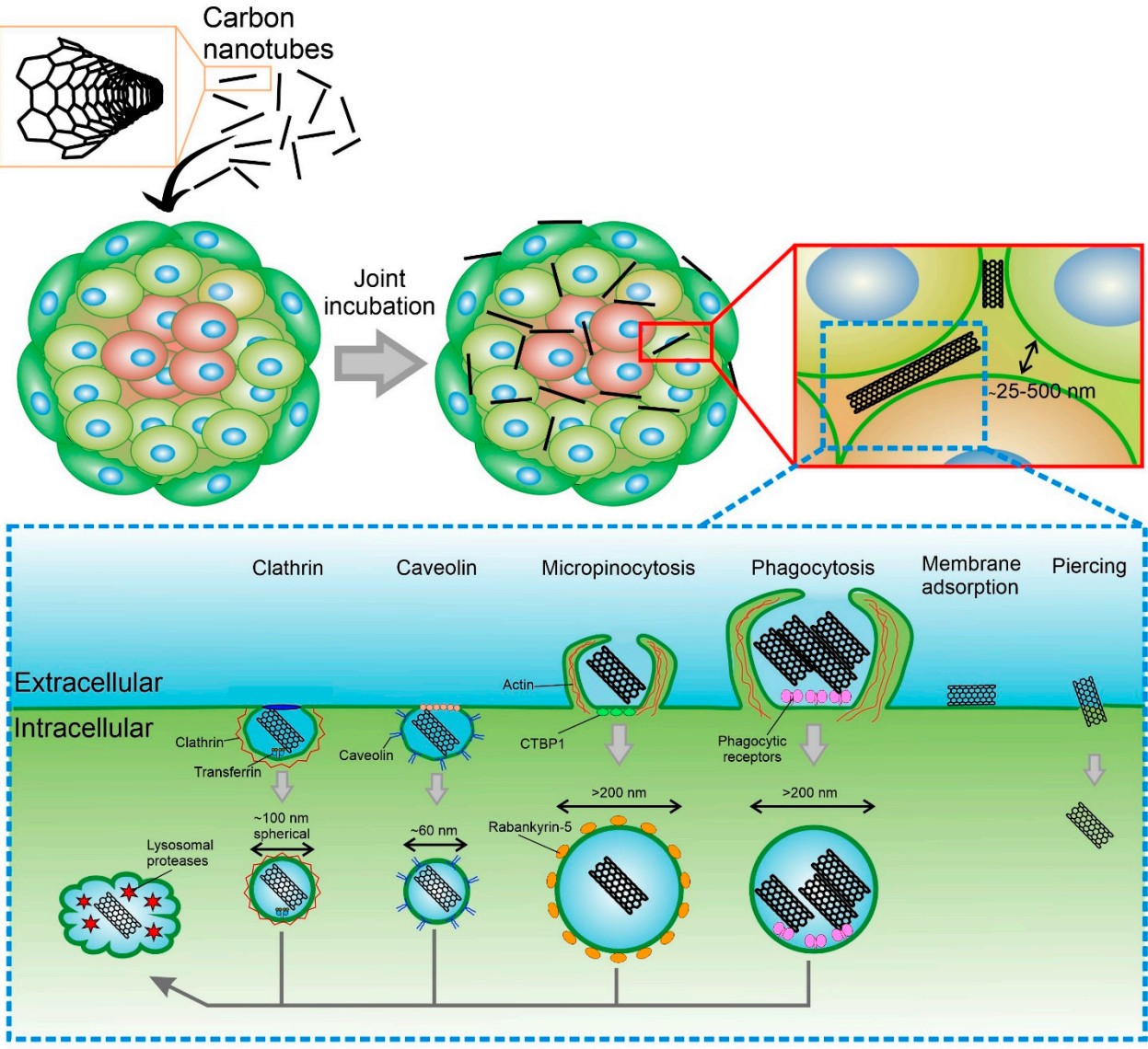

**Figure 3.** Overview of the primary mechanisms of uptake CNTs into cellular spheroid and cell.

There is now agreement in the literature that smaller particles penetrate spheroids faster. This has been tested with particles and spheroids of various types. CNTs are highly anisotropic objects with diameters ranging from ~0.4 to ~100 nm and lengths from ~0.14 nm to ~55 cm, so it is difficult to compare them with existing models. However, it is known that when 50 and 100 nm gold NPs penetrate for 24 h, 50 nm NPs penetrate deeper [104]. Similar results depending on the size could be obtained when the spheroids were immobilized in the "tumor-on-chip" system. This system made it possible to analyze the penetration of NPs in combination with real-time observation of the accumulation of NPs. Small spherical PEG-coated NPs (40 and 70 nm) rapidly accumulated in MDA-MB-435 spheroids and accumulated in the interstitial space, while larger NPs (110 and 150 nm) were more and more rejected from accumulation in the tumor [105]. Although the rule is clear that smaller particles have better penetration, no clear upper limit has been reported so far that would lead to the complete exclusion of particles from spheroid models, although there are indications that penetration becomes low after sizes larger than 1000 nm.

The next important feature is the shape of the particles. As mentioned earlier, CNTs are highly anisotropic particles. However, it has been previously repeatedly demonstrated that elongated small particles enter 2D cell culture much better than spheres. The results of this study show that the rate of internalization increases as the aspect ratio increases. If an

equal number of particles are added per cell, then the total volume of internalized particles increases with the volume of individual particles [106]. However, as we said above, a 2D system is very different from a 3D. Jiacheng Zhao et al. in their work describe particles from poly(1-O-methacryloyl-β-d-fructopyranose)-b-poly(methyl methacrylate) having the shape of spheres (diameter 30 nm), rods (diameter 30, length 120), and carriers (hollow sphere 160 nm in diameter). The study showed that there is no difference between the passage of spheres and rods into the spheroids, and both types of particles enter the spheroid at the same speed, unlike carriers [107].

CNTs can be internalized both by the outer layers of cells and by cells that are closer to the center of the spheroid. When NPs are ingested by cells, including CNTs, there are several types of internalization: active (energy-dependent), passive (energy-independent), and diffusion. The active pathway of CNTs' internalization through the cell membrane occurs by endocytosis. In the case of endocytosis, CNTs enter cells inside vesicles (endosomes), and then they are gradually transported to the perinucleolar space, becoming lysosomes [108]. Studies related to the selective inhibition of endocytosis pathways showed that CNTs internalization includes several pathways, such as macropinocytosis, caveolae-mediated endocytosis, and clathrin-dependent endocytosis [109]. The results show that macropinocytosis is the main mechanism of internalization of SWCNTs, while clathrin-mediated endocytosis is length-dependent and relatively important for the shortest CNTs. Phagocytosis allows the uptake of CNTs longer than 1 μm and conglomerates, as well as microsized composite particles with CNTs embedded in their structure [110,111]. When cells were incubated with CNTs at 4 °C, the internalization of particles was strongly reduced because low-temperature blocks all types of endocytosis. It is also known that the contact of CNTs with the cell membrane occurs from the tip [112]. For nanotubes with end caps or a carbon sheath at the ends, the uptake process involves tip recognition via receptor binding, rotation induced by asymmetric elastic deformation at the tube-bilayer interface, and finally, penetration into the cell in a nearly vertical direction. For nanotubes without caps and sheaths on their ends, the needle entry mode is not realized.

Passive diffusion of CNTs is not dependent on temperature or endocytosis, as the particles simply penetrate through the lipid bilayer [113]. It is already known that CNTs functionalized with amino acids can easily penetrate the cell without entering the lysosome. Removal of CNTs involves processes of exocytosis and enzymatic degradation. It has been reported that CNTs are displaced from cells by exocytosis several hours or days after internalization [113].

## 5. Study of CNTs internalization into CSs

There are several works where researchers determined the possibility of carbon tubes passing inside spheroids (Table 1). Prakrit V. Jena et al. evaluated the passage of CNTs into tumor spheroids of the SK-136 cell line derived from an orthotopic model of liver cancer and the MCF-7 cell line [114]. To determine their position inside the spheroid, they used the radiation of semiconductor CNTs in the near-infrared zone. The researchers determined that the penetration of CNTs into the interior of SK-136 spheroids is significantly faster than that into MCF-7 (Figure 4a,b). First of all, scientists attribute this to differences in the ECM and interstitial space. These parameters of the spheroids were evaluated using SEM and histological staining. The surface of the SK-136 spheroids has a dense fibrous structure, while the MCF-7 spheres had short protrusions. Similarly, SK-136 cells appeared to be closely fused, while MCF-7 cells were distinct. Histological sections show that SK-136 cells are densely packed together, while MCF-7 spheroids show less intercellular density and 10 times more intercellular space (Figure 4c). This work directly shows the relationship between the ability of carbon tubes to penetrate deep into tissues and the properties of the tissue itself.

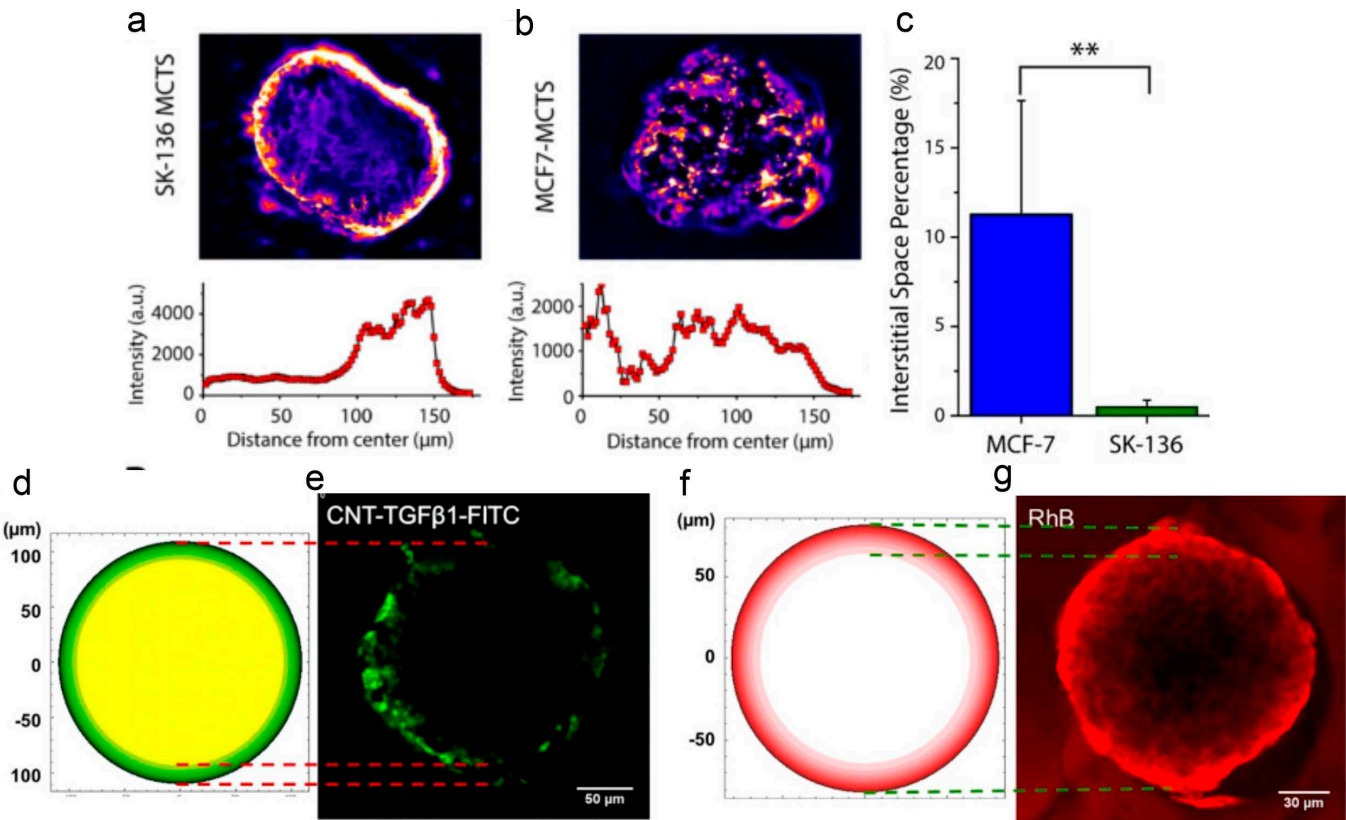

**Figure 4.** (**a**,**b**) Broadband fluorescence images through the central slices (midpoint of the MCTS in terms of *z*-axis distance) of (**a**) SK-136 spheroids and (**b**) MCF-7 spheroids. (**c**) Quantification of interstitial space percentage for MCF-7 and SK-136 MCTS. *p*-Value calculated by a two-sample t-test, ** indicates *p* < 0.01, n = 5 for each cell type. Reprinted with permission from ref. [114], 2016, Elsevier. (**d**–**g**) Calculated diffusion profile of (**d**) CNT-TGFβ1-FITC and (**f**) RhB, which compare to (**e**,**g**), permeation profiles of CNT-TGFβ1-FITC and RhB at central focal plane of the spheroid. Reprinted with permission from ref. [102], 2015, ACS.

Yichun Wang et al. also note the importance of using spheroids in their work [102]. The researchers argue that not only the properties of the NPs are important for good tissue penetration, but also the properties of the tissues themselves. However, the penetration profiles of CNTs obtained from the dissection of animal organs are difficult to analyze in terms of transport mechanisms due to the limited number of time points, natural variability of animals, and blood circulation. In this case, it is most convenient to use 3D CSs. First, the authors estimated the apparent diffusion coefficients for FITC, TGFβ1, CNT-FITC, and CNT-TGFβ1-FITC. TGFβ1 was chosen as the targeting ligand because TGFβ receptors are present in HepG2 cells. The researchers calculated experimental diffusion profiles according to Fick's second law using their code. However, it has been observed that CNTs-FITC have the same diffusion rate as free FITC (Figure 4d–g). Such a significant acceleration of large particles such as CNTs raises a lot of questions. For explanation, the authors resorted to scanning electron microscopy, histological sections, and theoretical calculations. Thus, the abnormally high values of diffusion of a CNTs-TGFβ1-FITC should be explained by the contribution of lateral diffusion along the cell surface to the total transport. The electrostatic repulsion between CNT-TGFβ1-FITC and the cell membrane facilitates lateral movement similar to sliding. Partial surface retention due to the presence of targeting ligands dramatically accelerates permeant transport despite an overall increase in mass and results in abnormally high diffusion coefficients. A similar transition from 3D to 2D diffusion in tissues is also known for some proteins that roll across the cell membrane. The method described in that paper provides an accurate and systematic evaluation of

various CNT transport modes and CNT-based drug delivery systems required for complex pharmacokinetic models.

**Table 1.** Representative examples of the research study of carbon nanotubes on 3d CSs.

| Aim of Study | CNTs Properties | | | Spheroids Properties | | | Effect | Ref. |
|---|---|---|---|---|---|---|---|---|
| | Structure | Diameter/Length | Charge (mV) | Cell Type | Spheroid Diameter | Number of Cells/ Spheroid | | |
| **Internalization** | SDC-SWCNT | -/232 nm | - | SK-136, MCF-7 | ∼100 μm | 500 | For SK-136 penetration into the surface cell layer; For MCF-7 penetration into the center of the spheroid. | [114] |
| | CNTs-TGFβ1-FITC | 1.2 nm/ 1000 nm | −8.4 ± 0.31 | HepG2 | 141.9 ± 5.6 μm | - | 20 μm penetration after 20 min of joint incubation. | [102] |
| **Drug Delivery** | SWCNTs-DOX-HA | 1–2 nm/ 1–3 μm, | 55.73 ± 0.89 | MDA-MB-231 | - | 5000 | Penetration into center of spheroid. After five days of joint incubation, spheroids broke because of cell apoptosis. | [115] |
| | TBMWCNTs@OXA | -/∼1 μm | +25.9 | U87 | - | 4000 | Penetrated throughout the interior of the spheroids and were detected at depths of over 100 μm. Laser exposure stopped the growth of spheroids and their fusion. | [116] |
| | EPI-SWCNTs-DSPE-HA | -/179.42 ± 1.96 nm | −47.6 ± 2.64 | A549 | /∼100 μm | 500 | 75% reduction in spheroid volume after six days of co-incubation. | [117] |
| | CNT-DOX | - | −13.9 ± 0.67 | HT29 | - | 50,000 | Incubation CNT-Dox at concentrations of 20.0 to 1.25 μg/mL with trypsin at a concentration of 0 to 70% led to a dose-dependent decreasing the percentage of living cells from 80.9 and 99.8%, respectively. | [118] |
| | SAL-SWNTs-CHI-HA | 1–2 nm/ 5–20 μm | −11.23 ± 1.15 | AGS cells | - | 10,000 | Significantly decreased the proportion of CD44+ cells, the ability of mammosphere and colony formation, and the growth of gastric CSC mammosphere. | [119] |
| **Photothermal Therapy** | Pab−MWCNTs | -/0.5−2 μm | −18 ± 1.4 | NCI/ADR-RES | - | 8000 | Penetrated into spheroid and produced cancer cell death after laser irritation. | [120] |
| | MWCNTs-DSPE-PEG | 8–15 nm/ 0.5–2 μm | −27.9 ± 0.4 | U87 | - | 4000 | Penetrated throughout the interior of the spheroids and were detected at depths of over 100 μm. Laser exposure stopped the growth of spheroids and their fusion. | [121] |
| | SWNT/chitosan-anti-CD133-PE | -/233 nm | +40 | CD133+ cells from GBM tissues of patients | - | 10,000 | After laser exposure, cell migration from the spheroid significantly decreased. | [122] |
| | Ru@SWCNT | ∼0.7–1.3 nm/From 20 nm to several micrometers | - | HeLa | ∼400 μm | 6000 | After 5 min of laser irradiation, the cell viabilities of the MCTSs were only 5%. | [123] |
| **Other** | PLLA/MWCNT | - | - | HBMC | - | 15,000 | HBMC/An increased osteocalcin expression. | [124] |
| | MWCNTs | 5–15 nm/ 0.5–2 μm | - | iPSCs DYR0100 | ∼1.5 mm | - | MWCNTs induced cytotoxicity and reduced NO-nNOS levels in 3D brain organoids. | [125] |
| | SWCNTs | 1.5 nm/ 1–5 μm | −9.96 ± 0.42 | Stem Cells | - | 30,000 | SWCNTs induced stem cell properties by spheroid formation, anoikis/apoptosis resistance, and stem cell markers expression. | [126] |

## 6. Study of CNTs as A Drug Delivery System on CSs

Carbon nanotubes are widely used as targeted delivery systems. New ways of treating cancer, based on carbon nanotubes, are being developed. Thus, by synthesizing different complexes, it is possible to achieve a reduction of negative effects on the entire body.

In a recent paper, the authors used Hyaluronic acid (HA)-modified amino single-walled to target DOX. SWCNTs-DOX-HA complexes with a high DOX loading were formed [115]. In vitro study showed that the release of DOX was faster at low pH values of 5.5, which corresponds to the tumor cell microenvironment, than in physiological conditions at pH 7.4. The rate of DOX release from SWCNTs-DOX-HA complexes is lower than that of SWCNTs-DOX. The authors used the MDA-MB-231 line as an example; it was shown that SWCNTs-DOX-HA complexes suppressed cell proliferation and induced apoptosis better than unmodified SWCNTs-DOX. In the cancer cell spheroid assay, SWCNTs-DOX-HA demonstrated a marked effect of inhibiting cancer cell spheroid growth. DOX was retained on SWCNTs and reduced toxic and side effects on normal cells.

Perepelytsina, O.M. et al. studied the toxicity of oxidized carbon nanotubes (CNTox) functionalized with doxorubicin (CNT-Dox) on tumor cells in vitro (2-D, 3-D cultures) and on Balb2/c mice models in vivo [118]. The possibility of immobilization and subsequent release of DOX from the CNT surface was shown, as well as a decrease in cytotoxicity of CNT-DOX compared to DOX. The combined use of CNT and DOX after release allows for greater efficacy in suppressing tumor growth in vitro. In 3D culture, increasing in CNTs concentration is accompanied by a dose-dependent increase in the median volume of spheroids. In vivo was studied the effect of the obtained structures on the state of the hepatic enzymatic system, the protein metabolism, and cell blood composition of the mouse. CNT-DOX showed less overall toxic interaction on the body compared to pure DOX.

Yao, H.J. et al. synthesized single-walled carbon nanotubes (SWCNTs) distearoylpho sphatidylethanolamine-hyaluronic acid (DSPE-HA) with a single coupling point, to simultaneously disperse SWCNTs, to improve the biocompatibility of SWCNTs, and target SWCNTs to CD44-overexpressing (Figure 5a) [117]. The authors used epirubicin (EPI) as a model drug and functionalized DSPE-HA SWCNTs as a carrier, and created EPI-SWCNTsDSPE-HA drug delivery systems. The efficacy of delivery of EPI-SWCNTsDSPE-HA complexes on A549/Taxol cells and tumor spheroids was investigated (Figure 5b). They showed that EPI-SWCNTs-DSPE-HA significantly promoted the intracellular accumulation of EPI in multidrug resistance cancer cells via CD44 receptor-mediated endocytosis.

Yao, H. et al. used complexes based on chitosan coated with single-walled carbon tubes with salinomycin (SAL) functionalized with hyaluronic acid (HA) as targeted delivery systems (SAL-SWNTCHI-HA) (Figure 5c) [119]. SAL-SWNT-CHI-HA complexes are capable of inhibiting the self-renewal capacity of the CD44+ population and decreasing mammosphere- and colon-formation of gastric cancer stem cells. In addition, migration and invasion of gastric CSCs were significantly blocked by SAL-SWNT-CHI-HA complexes. SAL-SWNTs-CHI-HA significantly reduced the proportion of CD44þ cells, the ability to form mammosphere and colonies (Figure 5d), and the growth of gastric cancer stem cells. Additionally, the migration and invasion of gastric cancer stem cells were significantly inhibited by SAL-SWNTs-CHI-HA.

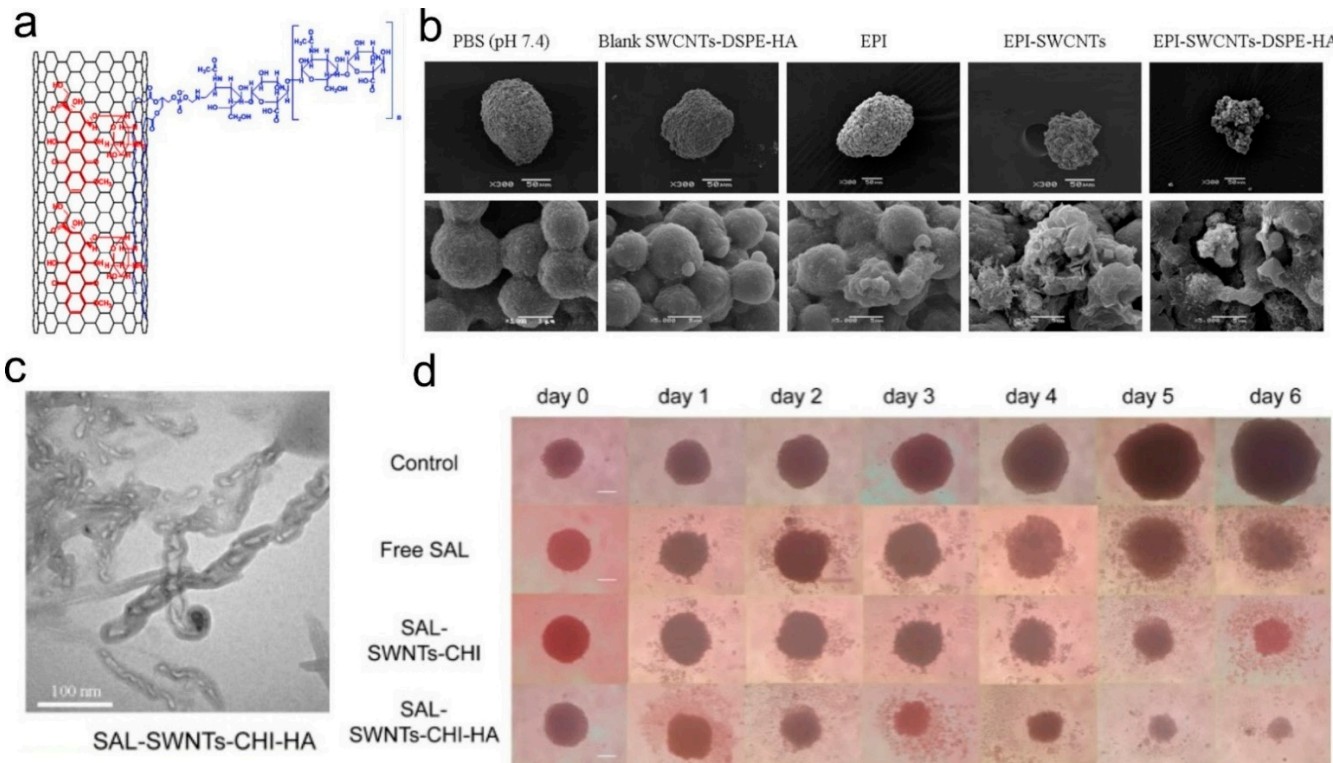

**Figure 5.** (**a**) Schematic representation of EPI-SWCNTs-DSPE-HA. (**b**) The surface morphology of A549/Taxol tumor spheroids after treatment with different formulations visualized by SEM on day 3. The first and second lines show full (×300) and magnified images (×5000) of the spheroid. Reprinted with permission from Elsevier [117]. (**c**) Transmission electron microscopy (TEM) image of functionalized SWNTs. Scale bar = 200 nm. (**d**) The CSCs mammospheres images treated with different SAL-containing formulations under an inverted microscope. The scale bar was 100 μm. Reprinted with permission from Elsevier [119].

## 7. Study of CNTs as an NPs for Photothermal Therapy on CSs

Multiple drug resistance (MDR) with P-glycoprotein (Pgp) remains a major challenge for cancer treatment. Since traditional approaches using low molecular weight inhibitors have failed in clinical development due to a lack of cancer specificity, anti-cancer researchers are developing carbon nanotubes targeting Pgp to achieve highly cancer-specific therapy by combining antibody-based cancer targeting and local tumor ablation with photothermal therapy. Refs. [127–129] Suo, X. et al. used antibody-based delivery systems, Pgp-specific antibodies (Pab-MWCNTs) (Figure 6a) [120]. They propose this as one option to solve the problem of medicated multidrug resistance. Pab-MWCNTs complexes showed high photocytotoxicity in multispheroids NCI/ADR-RES one day after laser irradiation (970 nm; 6 W/cm$^2$; 45 s) (Figure 6b).

Eldridge, B.N et al. the authors propose to use the property of NIR absorption and subsequent heat release by carbon nanotubes to develop new ways to treat glioblastoma multiforme [121]. They used DSPE-PEG MWCNTs structures that allow improved diffusion through brain phantoms while retaining the ability to reach ablative temperatures after laser exposure (Figure 6c). The authors internalized DSPE-PEG MWCNTs complexes into glioblastoma U87 cell lines and irradiated them with a laser (970 nm; 3 W/cm$^2$; 90 s) (Figure 6d). Laser exposure stopped the growth of spheroids and their fusion into spheroids.

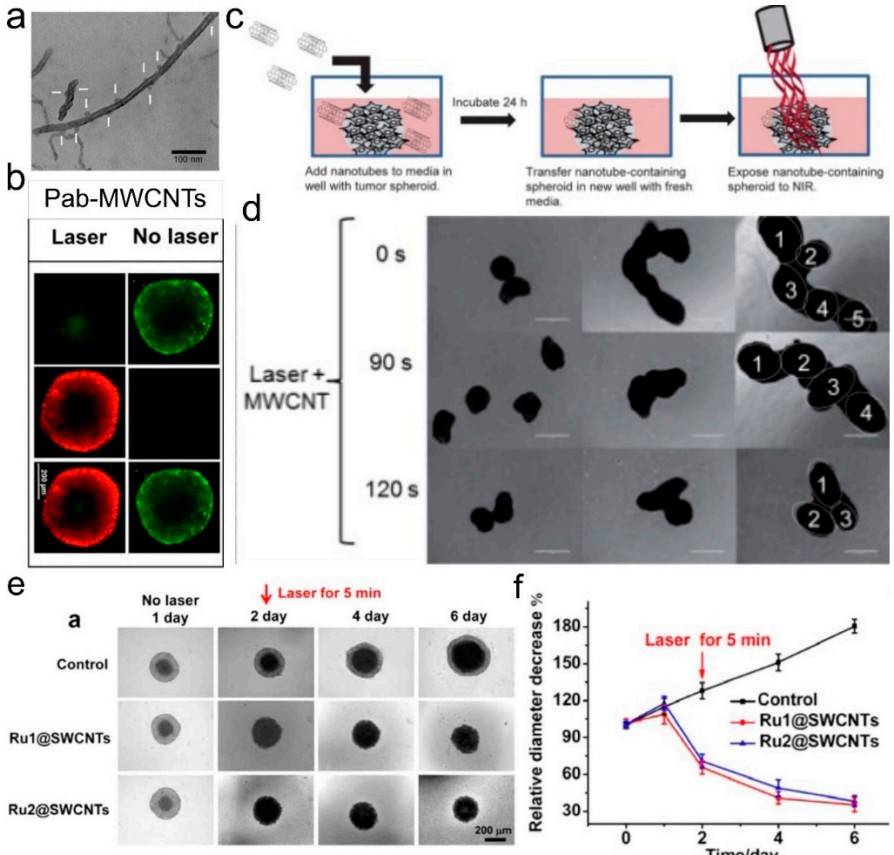

**Figure 6.** (**a**) Photoelectron micrographs of Pab–MWCNTs. White arrows indicate the location of antibody molecules on MWCNTs. (**b**) Live/dead cell staining with Calcein AM and PI in NCI-ADR/RES spheroids was performed one day after irradiation. Reprinted with permission from ref. [120], 2018, ACS. (**c**) Schematic illustrating the experimental design. (**d**) Spheroid growth over time was monitored, and representative photomicrographs are shown. Reprinted with permission from ref. [121], 2016, ACS. (**e**,**f**) Diameter change of Ru@SWCNTs and irradiation incubated with tumor spheroids by increasing days. Reprinted with permission from ref. [123], 2015, ACS.

Zhang, P. et al. the authors used Ru(II) complex-functionalized single-walled carbon nanotubes (Ru@SWCNTs) as nano templates for bimodal photothermal and two-photon photodynamic therapy (PTT-TPPDT) [123]. After laser treatment (808 nm; 0.25 W/cm$^2$; 5 min), Ru(II) complexes were released from Ru@SWCNTs via photothermal triggers. HeLa spheroids with Ru@SWCNTs were irradiated with laser in the same mode, and a significant decrease in cell viability (up to 5%) was shown (Figure 6e,f).

In a recent paper by Wang, C.H. et al., GBM-CD133+ and GBM-CD133− cells were exposed to single-walled carbon nanotubes (SWNTs) conjugated with the monoclonal antibody CD133 (anti-CD133) and then irradiated with near-infrared laser light [122]. The results showed that GBM-CD133+ cells were selectively targeted and killed, while GBM-CD133 cells remained viable. Moreover, the tumorigenicity and self-renewal ability of GBM-CD133+ cells treated with localized hyperthermia was significantly blocked. Furthermore, GBM-CD133+ cells pretreated with anti-CD133-SWNTs and irradiated with a near-infrared laser 2 days after xenotransplantation to nude mice did not exhibit stable cancer stem-like cell signatures for tumor growth.

## 8. Additional Possibilities for the Application of CNTs on CSs

The use of CNTs for tissue engineering purposes also holds great promise since they can significantly improve the mechanical and conductive properties of tissues, which is required, for example, in the engineering of bone and nerve tissue. [38,130] CNTs can also

be used to create tissue-engineered structures that represent models of various diseases. For example, recent studies by Kiratipaiboon et al. showed that CNT exposure could transform normal human lung fibroblasts (NHLFs) toward stem cells or stem-like cells. These fibroblast-associated stem cells (FSCs) are capable of forming collagen-rich fibroblastic foci similar to those noticed in animal models and patients with pulmonary fibrosis. In patients, the formation of fibroblastic foci has commonly been used as a reliable marker of poor prognosis. Such structures have also been shown to be induced by CNTs. They contain high levels of collagen and stem cell markers such as aldehyde dehydrogenase (ALDH) activity, ATP binding cassette subfamily G member 2 (ABCG2), and CD90 based on our previous studies. Animal studies also showed the overexpression of these stem cell-related markers in fibrotic lesions of CNT-exposed lungs. Together, these studies suggest the putative role of FSCs in CNT-mediated fibrosis [126].

Jiang et al. showed that MWCNTs induced cytotoxicity and reduced NO-nNOS levels in 3D brain organoids. As a possible mechanism, exposure to MWCNTs altered the protein levels of nNOS regulators NF-κB and KLF4. Most importantly, the images obtained by the fluorescence MOST method indicated that the decrease in nNOS proteins occurred not only at the out-layers but also the inner-layers of 3D brain organoids, which suggested that MWCNTs could effectively influence the whole organoids, which are multi-layered. As 3D brain organoids derived from human iPS cells resemble human brains, it is expected that the data obtained from 3D brain organoids could be better extrapolated to humans compared with non-human-based models. Thus, 3D brain organoids could be applied as an advanced in vitro platform to investigate the neurotoxicity of MWCNTs [125].

## 9. Conclusions

In general, working with 3D cellular structures, and CSs in particular, is a multidisciplinary field and has the potential for explosive growth. In this study, we discussed how CSs could be a tool for CNTs research, their internalization in 3D cellular structures, use as carriers for the delivery of therapeutic molecules, and photothermal therapy. We summarized the parameters that directly affect the application of CNTs in nanobiomedicine. In particular, we considered the size, number of walls, charge, and surface modifications that were evaluated as the most important. We also reviewed the most important parameters of 3D spheroids as opposed to 2D cell cultures and methods of fabrication of the most optimal spheroids, which is most important for groups who are just starting to work with 3D cell cultures. We described the mechanism of CNTs penetration into cellular spheroids and the molecular mechanisms of particle uptake by individual cells. The works describing the process of penetration of CNTs inside spheroids and, as a consequence, the works describing the process of studying CNTs as carriers of therapeutic molecules and their use as a platform for photothermal therapy have been described in most detail. There is no doubt that CSs can provide valuable information, such as information on the penetration and effect of CNTs, which 2D models cannot make available. We are still at the stage where spheroids cannot replace in vivo studies, but the development of more complex models co-cultured with other cells that, for example, simulate angiogenesis and tumor fibrosis could bring us closer to reducing the use of animals in preclinical studies.

**Author Contributions:** Writing—original draft preparation, A.A.A. and R.A.A.; writing—review and editing, A.A.A. and D.A.G. All authors have read and agreed to the published version of the manuscript.

**Funding:** This research was funded by a grant from the President of the Russian Federation for junior postdocs (No. MK-933.2022.3).

**Data Availability Statement:** All data presented here are adopted from the published work cited in the references.

**Conflicts of Interest:** The authors declare no conflict of interest.

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
