# Peer review of "3D Cell Spheroids as a Tool for Evaluating the Effectiveness of Carbon Nanotubes as a Drug Delivery and Photothermal Therapy Agents"

_carbon, 2022_

Round 1

Reviewer 1 Report

In this manuscript, the authors discuss the current status of 3D culture testing methods in carbon nanotubes applications such as drug delivery and photothermal therapy. Although this whole work illustrated the authors’ objectives, several major issues need to be addressed before its publication. The following are some detailed comments:

 1. The authors have reviewed the research of 3D cell spheres in carbon nanohorn, but it is necessary to classify and discuss the necessity and importance, which is helpful to attract more attention.

 2. The author should carefully manage the logic and hierarchy of the whole text according to the conclusion section so that the article is easy to understand and has a clear logical structure.

 3. 3D cell spheres can mimic the specificity of native tissues better than cells cultured in monolayers by replicating the physiological interactions between cells and extracellular matrix, but it is currently not a substitute for in vivo studies. Therefore, the authors should explore different aspects of how to optimize this 3D culture detection methods and the challenges of its application in biomedicine.

 4. Some important references are suggested to be cited: Carbon, 2019, 143, 814-827. Journal of Colloid and Interface Science,2022, 628, 273-286

 5. Some mistakes were found in the format and grammar. Please check and correct them carefully.

Author Response

  1. The authors have reviewed the research of 3D cell spheres in carbon nanohorn, but it is necessary to classify and discuss the necessity and importance, which is helpful to attract more attention.

Thanks for your comment, we have expanded the section "2. Properties, modifications and application of CNTs" by describing what other carbon basic materials exist and their applications in biomedicine. Relevant literature references have also been added.

  1. The author should carefully manage the logic and hierarchy of the whole text according to the conclusion section so that the article is easy to understand and has a clear logical structure.

Thank you for your comment. We noticed inaccuracies in the section names, and to improve the reader's perception and correct errors, we changed the section names.

  1. Introduction
  2. Properties, modifications and application of CNTs
  3. Properties, fabrication and application of CSs
  4. Mechanism of CNTs uptake by cells and spheroids
  5. Study of CNTs penetration into CSs
  6. Study of CNTs as a drug delivery system on CSs
  7. Study of CNTs as an NPs for photothermal therapy on CSs
  8. Conclusions

Also, the text of the conclusions was changed to improve the reader's perception.

  1. 3D cell spheres can mimic the specificity of native tissues better than cells cultured in monolayers by replicating the physiological interactions between cells and extracellular matrix, but it is currently not a substitute for in vivo studies. Therefore, the authors should explore different aspects of how to optimize this 3D culture detection methods and the challenges of its application in biomedicine.

Thank you very much for your valuable comment. You are right, the review actually lacks a section that describes the protocol for optimizing cellular spheroids. We have added text to section 3. Properties, fabrication and application of CSs

Optimizing spheroids for nanoparticle testing, in particular CNTs, is one of the important aspects of working with 3D cultures. [86] The cells used in the experiments should be in culture from 1 to 20 passages. The cells should be kept in a humidified incubator at 37°C and 5% CO2. Standard culture medium should be used for cultivation. Cells should have a 70-80% fill rate. Cultures should be transplanted with trypsin/EDTA solution (0.05% (wt/vol) trypsin and 0.02% (wt/vol) EDTA). There are several methods for creating cell spheroids: the hanging drop method,[87] ultra non-adhesive well plates,[88] magnetic nanoparticles,[89] incubation in hydrogels,[90] and the use of bioreactors. For CNTs testing, the first three methods are the most optimal, since they are the easiest to use in all laboratories, as well as have the smallest variation in the size of the spheroids. To create a spheroid 400 µm in diameter on the fourth day of formation, the desired concentration and cell proliferation rate must be determined. For this purpose, spheroids are formed from different numbers of cells (from 250 to 3000 cells/spheroid in the case of ultra non-adhesive plates, magnetic nanoparticles, suspended droplet method). When creating spheroids, it is recommended to use a multichannel pipette, which will reduce the standard deviation among the spheroid diameters to 5% in one plate and 10% in different experiments. Cells form a spheroid within 96 hours in a CO2 incubator at 37°C. A phase-contrast microscope with 5x and 10x lenses is used to determine the size of the spheroid. The microscope is used to assess the integrity, diameter, volume and roundness of the spheroid. Once the optimal number of cells has been determined, it is possible to proceed with CNTs testing. This requires titration of CNTs and making 2x solutions of the substances tested. After that, 50 µl of medium must be removed from the plate where the spheroids are formed and 50 µl of the test solution must be added, thus making the concentration of CNTs of the desired concentration. The spheroids can then be incubated with the CNTs for the desired time.

  1. Some important references are suggested to be cited: Carbon, 2019, 143, 814-827. Journal of Colloid and Interface Science,2022, 628, 273-286

Thank you for your comment, we have included links to these works in our article.

  1. Some mistakes were found in the format and grammar. Please check and correct them carefully.

Thank you for your comment, we will carefully check the manuscript for grammatical errors

Reviewer 2 Report

Overall: Nice work. Which fits the scope of the journal. I suggest the authors to do the following minor revisions:

1- add a part about toxicity of CNTs and the risk in the blood (circulation)? What is the size of CNTs appropriate to be tested in spheroid (3D) cells? Please specify in the main text

2- go through the entire manuscript to correct any typos and grammatical mistakes;

3- add a part about CNTs, transporters-based resistance in spheroid cells.

4. Highlight the interest to review CNTs compared to other C Nanomaterials like graphene.

5- can you add a part about tissue engineering, biosensing  and use of CNTs ?

6- make sure that the figures have benefited of copyright permission

7- I suggest you to cite these articles:

about C materials in biosensing 

Menaa F, Fatemeh Y, Vashist SK, Iqbal H, Sharts ON, Menaa B. Graphene, an Interesting Nanocarbon Allotrope for Biosensing Applications: Advances, Insights, and Prospects. Biomed Eng Comput Biol. 2021 Feb 24;12:1179597220983821. doi: 10.1177/1179597220983821. PMID: 33716517; PMCID: PMC7917420.

About C materials in tissue engineering 

Menaa F, Abdelghani A, Menaa B. Graphene nanomaterials as biocompatible and conductive scaffolds for stem cells: impact for tissue engineering and regenerative medicine. J Tissue Eng Regen Med. 2015 Dec;9(12):1321-38. doi: 10.1002/term.1910. Epub 2014 Jun 11. PMID: 24917559.

About transporters in spheroid cells 

Houben R, Wischhusen J, Menaa F, Synwoldt P, Schrama D, Bröcker EB, Becker JC. Melanoma stem cells: targets for successful therapy? J Dtsch Dermatol Ges. 2008 Jul;6(7):541-6. English, German. doi: 10.1111/j.1610-0387.2008.06786.x. Epub 2008 May 30. PMID: 18513214.

best,

the reviewer 

Author Response

Overall: Nice work. Which fits the scope of the journal. I suggest the authors to do the following minor revisions:

  • add a part about toxicity of CNTs and the risk in the blood (circulation)? What is the size of CNTs appropriate to be tested in spheroid (3D) cells? Please specify in the main text

Thanks for your valuable advice, we have expanded the section "2. Properties, modifications and application of CNTs" with information about the types of toxicity of CNTs.

  • go through the entire manuscript to correct any typos and grammatical mistakes;

Thank you for your comment, we will check the text again for grammatical mistakes

  • add a part about CNTs, transporters-based resistance in spheroid cells.

Thank you for your valuable comment, in our review we have already used articles where the drug resistance cell line was used. We have expanded the section "Study of CNTs as an NPs for photothermal therapy on CSs".

We have also added references to the literature on the effects of CNTs on drug resistance cultures.

Curcio, M.; Farfalla, A.; Saletta, F.; Valli, E.; Pantuso, E.; Nicoletta, F.P.; Iemma, F.; Vittorio, O.; Cirillo, G. Functionalized Carbon Nanostructures Versus Drug Resistance: Promising Scenarios in Cancer Treatment. Molecules 2020, 25, 2102, doi:10.3390/molecules25092102.

Yao, Y.; Zhou, Y.; Liu, L.; Xu, Y.; Chen, Q.; Wang, Y.; Wu, S.; Deng, Y.; Zhang, J.; Shao, A. Nanoparticle-Based Drug Delivery in Cancer Therapy and Its Role in Overcoming Drug Resistance. Front. Mol. Biosci. 2020, 7, doi:10.3389/fmolb.2020.00193.

Qi, Y.; Yang, W.; Liu, S.; Han, F.; Wang, H.; Zhao, Y.; Zhou, Y.; Zhou, D. Cisplatin loaded multiwalled carbon nanotubes reverse drug resistance in NSCLC by inhibiting EMT. Cancer Cell Int. 2021, 21, 74, doi:10.1186/s12935-021-01771-9.

  • Highlight the interest to review CNTs compared to other C Nanomaterials like graphene.

Thank you very much for your question, we have expanded the section "Properties, modifications and application of CNTs" with information about graphene and fullerenes.

  • can you add a part about tissue engineering, biosensing  and use of CNTs ?

Thank you for your valuable comment. We have added a new section "8. Additional possibilities for the application of CNTs on CSs", as well as expanded Table 1.

  1. Additional possibilities for the application of CNTs on CSs

The use of CNTs for tissue engineering purposes also holds great promise, since they can significantly improve the mechanical and conductive properties of tissues, which is required, for example, in the engineering of bone and nerve tissue. CNTs can also be used to create tissue-engineered structures that represent models of various diseases. For example, recent studies by Kiratipaiboon et al. showed that CNT exposure could transform normal human lung fibroblasts (NHLFs) toward stem cells or stem-like cells. These fibroblast-associated stem cells (FSCs) are capable of forming collagen-rich fibroblastic foci similar to those noticed in animal models and patients with pulmonary fibrosis. In patients, the formation of fibroblastic foci has commonly been used as a reliable marker of poor prognosis. Such structures have also been shown to be induced by CNTs. They contain high levels of collagen and stem cell markers such as aldehyde dehydrogenase (ALDH) activity, ATP binding cassette subfamily G member 2 (ABCG2), and CD90 based on our previous studies. Animal studies also showed the overexpression of these stem cell-related markers in fibrotic lesions of CNT-exposed lungs. Together, these studies suggest the putative role of FSCs in CNT mediated fibrosis.[116]

Jiang et al. showed that MWCNTs induced cytotoxicity and reduced NO-nNOS levels in 3D brain organoids. As the possible mechanism, exposure to MWCNTs altered the protein levels of nNOS regulators NF-κB and KLF4. Most importantly, the images obtained by fluorescence MOST method indicated that the decrease of nNOS proteins occurred not only at the out-layers, but also the inner-layers of 3D brain organoids, which suggested that MWCNTs could effectively influence the whole organoids which are multi-layered. As 3D brain organoids derived from human iPS cells resemble human brains, it is expected that the data obtained from 3D brain organoids could be better extrapolated to humans compared with non-human based models. Thus, 3D brain organoids could be applied as advanced in vitro platform to investigate the neurotoxicity of MWCNTs.[117]

  • make sure that the figures have benefited of copyright permission

All of the figures used in this review have been authorized for reprinting

7- I suggest you to cite these articles:

about C materials in biosensing 

Menaa F, Fatemeh Y, Vashist SK, Iqbal H, Sharts ON, Menaa B. Graphene, an Interesting Nanocarbon Allotrope for Biosensing Applications: Advances, Insights, and Prospects. Biomed Eng Comput Biol. 2021 Feb 24;12:1179597220983821. doi: 10.1177/1179597220983821. PMID: 33716517; PMCID: PMC7917420.

About C materials in tissue engineering 

Menaa F, Abdelghani A, Menaa B. Graphene nanomaterials as biocompatible and conductive scaffolds for stem cells: impact for tissue engineering and regenerative medicine. J Tissue Eng Regen Med. 2015 Dec;9(12):1321-38. doi: 10.1002/term.1910. Epub 2014 Jun 11. PMID: 24917559.

About transporters in spheroid cells 

Houben R, Wischhusen J, Menaa F, Synwoldt P, Schrama D, Bröcker EB, Becker JC. Melanoma stem cells: targets for successful therapy? J Dtsch Dermatol Ges. 2008 Jul;6(7):541-6. English, German. doi: 10.1111/j.1610-0387.2008.06786.x. Epub 2008 May 30. PMID: 18513214.

Thank you for your valuable comment, we have referred to these works in the relevant sections

Round 2

Reviewer 1 Report

References and special symbols need to be carefully checked and corrected.